# Refining Transitive and Pseudo-Transitive Relations at Web Scale

Shuai Wang, Joe Raad, Peter Bloem, and Frank van Harmelen

Department of Computer Science, Vrije Universiteit Amsterdam, The Netherlands
{shuai.wang | j.raad | p.bloem | frank.van.harmelen}@vu.nl

**Abstract.** The publication of knowledge graphs on the Web in the form of RDF datasets, and the subsequent integration of such knowledge graphs are both essential to the idea of Linked Open Data. Combining such knowledge graphs can result in undesirable graph structures and even in logical inconsistencies. Refinement methods that can detect and repair such undesirable graph structures are therefore of crucial importance. Existing refinement methods for knowledge graphs are often domain-specific, are limited to single relations (e.g. owl:sameAs), or are limited in scale. We present a challenge consisting of a number of datasets of transitive and pseudo-transitive relations and hand-labeled gold standards, as well as baselines. We introduce an efficient web-scale knowledge graph refinement algorithm that works for such relations. Our algorithm analyses the graph structure, and allows the use of weighting schemes to heuristically determine which possibly erroneous edges should be removed to make the graph cycle free. When compared against general-purpose graph algorithms that perform the same task, our algorithm removes the least amount of edges to make the graph of transitive relations cycle-free while maintaining a better precision in identifying erroneous edges as measured against a human gold-standard.

## 1 Introduction

The central tenet of Linked Open Data is the publication and integration of RDF datasets. Such integration can result in logical inconsistencies or undesirable graph structures. For transitive relations, this can result in chains of relation instances forming complex nested cycles involving many entities across datasets in the corresponding graph. In practice, even logically valid cycles may have negative consequences. For example, a cycle of `rdfs:subClassOf` triples in an intended hierarchy enforces equality of all classes in the cycle, which may prevent algorithms such as query expansion from termination. To ensure data quality, refinement methods have been developed [13]. However, these methods often depend on domain-specific functionalities [7], or limited to a specific relation (e.g. owl:sameAs) [15, 18] or suffer from limited scalability [18]. Such limitations call for the development of scalable and domain-independent algorithms.

This paper presents a new approach for detecting undesirable cycles in transitive relations. It uses graph structural characteristics and a heuristic notion

of reliability of triples, without the need for any domain-dependent information such as labels, comments and other textual information in context [7]. Our approach (i) is independent from domain and language, (ii) has a better precision than general-purpose graph-theoretical methods and (iii) maintains good scalability and efficiency.

Graph structure reflects logical properties and vice versa. For example, when a relation is asymmetric, any cycle of size two in its graph violates consistency. Similarly, for irreflexive relations any self-loop is invalid. This suggests the use of graph-theoretic algorithms to detect logical inconsistencies. In OWL, the transitivity of a relation is typically specified directly through *owl:TransitiveProperty*. In this work, we extend to what we call *pseudo-transitive* relations: that of a subproperty or the inverse of a (pseudo-)transitive relation and those whose intended semantics is assumed to be both transitive and anti-symmetric, although not formally asserted. In this paper, we exclude equivalence relations (e.g. `owl:sameAs`) and those whose (pseudo-)transitivity are mistakenly asserted or implied on the LOD Cloud (e.g. `foaf:knows`).

Besides the graph structure, another feature to be used for the refinement of knowledge graphs that is independent from domain and language is the *reliability* of triples. While there can be different heuristics, we measure reliability of an edge by counting the number of the occurrences of this edge across datasets of the web-scale integrated graph (see more details in Section 5.2). For small self-sufficient datasets, this feature is not of great value because the logical foundations of knowledge graphs dictate that repeated statements in datasets are redundant. However, such feature has been shown to be useful for the ranking of documents and entities [6] and the identification of erroneous assertions and improvement of data quality [3] when the sources of data are present. Figure 1 is an example subgraph of `skos:broader` with such weights extracted from the LOD Laundromat 2015 crawl [1]. It is more likely that `dbc:Numbers` is a broader than `dbc:Integers` (weighting 72), while it is unlikely that `dbc:Integers` is a broader concept for `dbc:Numeral_systems` (weighting 1), showing that weights can indicate the reliability of edges. This example also shows that the relation between some entities can be ambiguous, making it difficult to construct a perfect gold standard. For example, some may believe that numbers are parts of numerical systems while others may think the study of numbers includes the study of numerical systems. Finally, it also indicates that the weights of edges in the neighbourhood can have an impact on the reliability of edges.

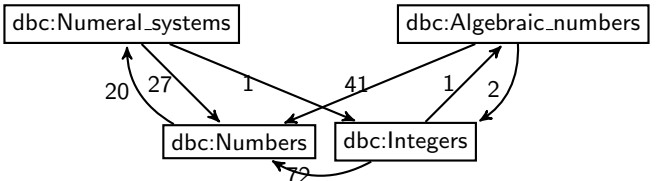

Fig. 1: An example subgraph of `skos:broader` with weights.

The hypotheses that we pursue in this paper are as follows:

H1: By taking graph structural properties (how edges are involved in complex nested cycles) into account, we can make knowledge graphs acyclic while removing fewer edges than graph theoretical methods.

H2: Taking the reliability of triples into account improves the accuracy for identifying erroneous edges.

This paper presents an algorithm to refine transitive and pseudo-transitive relations for large integrated knowledge graphs at web scale by removing as few edges as possible to obtain acyclic graphs. More specifically, the paper makes the following contributions:

1. a new metric for the hardness of resolving cycles based on strongly connected components.
2. a generic scalable approach for the refinement of (pseudo-)transitive relations using an SMT solver by exploiting Strongly Connected Components.
3. an evaluation that shows how taking into account the reliability of triples can improve the precision of the graph refinement algorithm.
4. a dataset of several widely used (pseudo-)transitive relations with their reliability weights.
5. a new gold standard of thousands of manually annotated triples to be used in the evaluation and comparison of graph refinement algorithms.

This paper is structured as follows: Section 2 and 3 discuss related work and present preliminaries. Section 4 describes the dataset and analyses its complexity. Section 5 presents our approach for refining (pseudo-)transitive relations. Section 6 presents the implementation details, our gold standard, and the conducted evaluation. Section 7 discusses the results and concludes the paper.

## 2    Related Work

### 2.1    Knowledge Graph Refinement Methods

According to Paulheim's survey [13], there are two main goals for knowledge graph refinement methods: *completing* the knowledge graph with missing knowledge, and *correcting* asserted information. This work falls into the latter category of approaches, as we aim at refining transitive and pseudo-transitive relations by removing edges that lead to unwanted cycles and are potentially erroneous. The closest predecessor of our work is the approach by [18], introduced for refining edges of `rdfs:subClassOf` by exhaustively listing simple cycles[1] and removing minimal edges so that the resulting graph is cycle free. However, this approach faces a combinatorial explosion when listing all simple cycles of large nested clusters, and therefore cannot be applied on some relations we study in this

---

[1] A simple cycle is a cycle in which the only repeated vertices are the first and last vertices.

work. Sun et al. [17] propose similar strategies for removing edges causing cycles in graph. However, this approach requires inferring a graph hierarchy (e.g. using a Bayesian skill rating system), and has been only tested on synthetic datasets and the Wikipedia category graph. Another recent approach that targets the refinement of categorical and list information is introduced by [7]. To our best understanding, this graph-based refinement approach relies on external information of hypernyms, and applies only to English DBpedia categories and lists. Moreover, and similarly to the work presented in [5], these approaches assume the existence of a hierarchy and takes advantage of pre-defined roots. In this work, we show that such hierarchies are frequently violated in the Web, therefore the applicability of such approaches becomes limited in such context. Finally, the graph-based approach presented in this work is similar to other approaches that have also exploited the graph structure for detecting and removing different types erroneous edges at the scale of the Web, such as type [14] and identity links [15].

## 2.2   General MWFAS Algorithms

In this section we discuss general-purpose graph algorithms for making graphs cycle-free. When restricting to a single relation, the problem of resolving cycles is identical to finding the *Maximum Weighted Directed Acyclic Subgraph* (MW-DAS).[2] Historically, the removed edges are also called *arcs* and form a *feedback arc set* (FAS). Therefore, the problem is equivalent to *Minimum Weighted Feedback Arc Set* (MWFAS), and we will use these names for the rest of the paper. The MWFAS problem is APX-hard. Despite the hard limit on its approximability, there are polynomial-time approximation algorithms. The following summarises some algorithms that scale to at least tens of millions of edges according to [16] where more details are presented.

The underlying idea of the **KwikSort(KS)** algorithm is to sort vertices on the number of back arcs induced, and removing the edges with many induced back arcs. The algorithm runs at $\mathcal{O}(n \log n)$ when assuming that arc membership can be tested in constant time. In our experiments we used an optimised implementation that uses $\mathcal{O}(n \log n)$ additional space. Since KS takes a random initial ordering, we take the best result of 200 runs.

The **Greedy(GRD)** algorithm greedily appends all "sink-like" vertices at the end of a sequence $s$ and inserts the "source-like" vertices at the front of $s$. The implementation in [16] uses bins [12] for the selection of vertices in each iteration. The bins distinguish nodes with only outgoing edges, nodes without outgoing edges, and nodes with both in- and outgoing edges. Each node falls in one of these bins. By using $s$ as a linear arrangement and picking all the feedback arcs, it minimize the number of arcs with different orientation. GRD runs in time $\mathcal{O}(m+n)$ and uses $\mathcal{O}(m+n)$ space. It has a guarantee of removing

---

[2] Note that the resulting graph may not be a spanning tree but a set of directed acyclic graphs (DAGs). Therefore this problem cannot be solved by minimum spanning tree algorithms.

no more than $\frac{1}{2}|E| - \frac{1}{6}|V|$ edges but experiments from [16] observed that the size of FAS is drastically smaller than this worst-case bound.

For a graph $G$, the **BergerShor(BS)** algorithm begins with a random permutation over the vertices $V$. It then processes each vertex by comparing its in-degree and out-degree. If a vertex has more incoming arcs than outgoing ones, the incoming ones are removed and added to a set $E'$ while the outgoing arcs are removed and discarded. The collected arcs $E'$ form an acyclic graph $G'$ (its counterpart is the set of arcs removed). The algorithm runs in time $\mathcal{O}(m+n)$ and [16] show that the algorithm far out-performs this worst-case bound.

Finally, we can adopt a **depth-first traversal(DFS)** algorithm and remove all arcs that form a cycle during the search to ensure that the resulting graph is acyclic. Its runtime complexity is $\mathcal{O}(m + n)$. The algorithm does not make any intelligent decision nor minimize the resulting size of FAS.

## 3    Preliminaries

A knowledge graph is a directed and labelled graph $G = \langle V, E, \Sigma_E, l_E \rangle$, where $V$ is the set of vertices (nodes), $E \subseteq V \times V$ the set of relations (edges), and $\Sigma_E$ is the set of edge labels. $l_E : E \to 2^{\Sigma_E}$ is a function that assigns to each edge in $E$ a set of labels belonging to $\Sigma_E$. For a specific relation $R \in \Sigma_E$, we denote $G_R = \langle V_R, E_R \rangle$ the edge-induced subgraph that only includes those edges whose labels are $R$, with $V_R \subseteq V$ and $E_R \subseteq E$. In the case of weighted graphs, we introduce an additional weight function $f_w : E \to \mathbb{N}$ that assigns to each edge a weight. In Section 5.2, we describe how these weights are calculated.

A walk in a graph $G_R$ is a sequence of vertices $v_0, v_1, \ldots, v_n$, with the edge $(v_i, v_{i+1}) \in E_R$. A walk is a *path* if no edge is repeated. A path between two vertices is *shortest* if the number of vertices on the path is minimal. A path is a *cycle* if it is closed, i.e. $v_0 = v_n$. Apart from reflexive relations, the smallest possible cycle involves two vertices. We denote the set of such size-two cycles by $E_{C2}$ for later convenience. A graph without the corresponding edges of $E_{C2}$ (and vertices if left as singletons) is denoted $G' = G \backslash E_{C2}$.

A Strongly Connected Component (SCC) of $G_R$ is a subgraph where any two of its vertices can be reached by a path (i.e. the subgraph is strongly connected) and is maximal for this property: no additional edges or vertices can be included in the subgraph without breaking strong connectivity.[3] The collection of all SCCs forms a new graph $G^{SCC}$, and the set of SCCs of $G$ is denoted $\kappa(G)$. In this paper, the process of removing edges in an SCC to make it acyclic is referred to as *resolving cycles*, and graphs with no cycles are referred to as DAGs (directed acyclic graphs). In our datasets, there is often one SCC that is significantly larger (with the most vertices and edges) than others among all the SCCs of a graph. We refer to it as $G^B$ when discussing its properties.

Figure 2 presents an example of a graph $G$ with its introduced variants: $G^{SCC}$, $G'$, and $G'^{SCC}$. Note that cycles of size two are not necessarily SCCs of

---

[3] All SCCs in this paper are assumed to have more than one node.

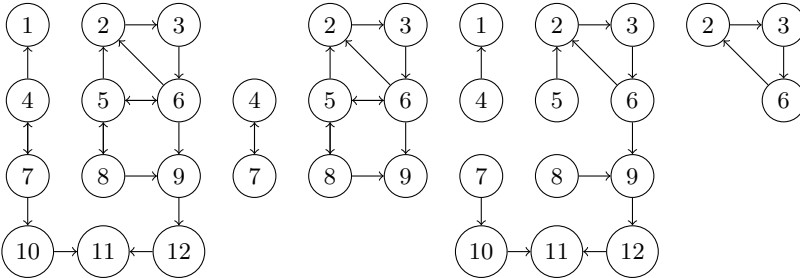

Fig. 2: An example graph and its variants (from left: $G$, $G^{SCC}$, $G'$, $G'^{SCC}$).

size two as they can be nested into other cycles and form a bigger SCC (e.g. size-two cycle between node 5 and 6).

There are efficient algorithms for computing the SCCs of a graph such as Tarjan, which take linear time $\mathcal{O}(|V|+|E|)$ assuming constant time for retrieving edges [9]. It is useful to observe that cycles in a graph $G$ can never span across multiple SCCs (since if there were any such cycle, the SCCs involved would form a bigger SCC, which contradicts its maximality w.r.t. strong connectivity). Therefore, since cycles in $G$ are always contained inside a single SCC, and since the collection of all SCCs of $G$ form a partition of the vertices of $G$, we can safely divide-and-conquer the process of resolving cycles in $G$ across all SCCs of $G$. This allow us to focus the cycle resolution locally in comparison with inefficiently and exhaustively listing all simple cycles as in [18].

## 4    Pseudo-Transitive Relations in the LOD Cloud

### 4.1    Dataset

In this work, we use the LOD-a-lot dataset [4] as a representative copy of the LOD Cloud. This compressed data file of 28 billion unique triples is the result of the integration of over 650K datasets that are crawled and cleaned by the LOD Laundromat in 2015 [1]. In the LOD-a-lot dataset, there are 2,486 relations explicitly stated as `owl:TransitiveProperty`, used in more than 776 million triples (2.7% of all triples). When the semantics of `rdfs:subPropertyOf` and `owl:inverseOf` is exploited, the number of (pseudo-)transitive relations increases to 8,687 relations, used in around 5.5 billion unique triples (19.5% of the triples). Our manual examination shows that a number of these properties are incorrectly asserted or inferred, such as the widely used `foaf:knows` relation.

For transitive relations, graph characteristics can reflect the logical properties, and vice versa. For instance, irreflexive and antisymmetric relations such as `iwwem:dependsOn` allow for no cycle anywhere in the graph. We consider `skos:broader` a pseudo-transitive relation, as it was not designed to be a transitive property despite being a subproperty of `skos:broaderTransitive`, which is typed `owl:TransitiveProperty` [10]. Unless otherwise specified, we assume

that the graph of relations such as `rdfs:subClassOf` and `geo:parentFeature` should be cycle-free despite the logical validity of cycles. Our maual examination also found that many relations are defined together with their inverse (e.g. `skos:broader` and `skos:narrower`). There can also be a relation like that of equivalence (e.g. `owl:sameAs, rdfs:equivalentClassOf`). This paper examines a selection of 10 relations (see e.g. Figure 3 and Table 1). These are popular relations, all of them directly typed as `owl:TransitiveProperty` with over 100k triples. We exclude the few that actually represent equivalence relations, or whose biggest SCC has less than 10 vertices unless its inverse is to be studied.

## 4.2 Strongly Connected Components Analysis

To get a sense of how difficult it is to make graphs cycle-free, we introduce in this section a number of metrics. We may turn to standard metrics for the degree of transitivity of a graph, such as the transitivity index $T$ (the number of actual triangles in a graphs as a fraction of the number of all possible triangles), the average clustering index $C$ (the average over the local clustering coefficients of all vertices, where a local clustering coefficient of a vertex is the actual number of edges in the direct neighbourhood of the vertex divided by the possible number of such edges), or the global reaching centrality ($GRC$) [11]. These measures can be useful for the understanding of graph-theoretical properties. However, our analysis shows that none of $T$, $C$ or $GRC$ manage to capture the size of SCCs or the hardness of cycle-resolution. Thus they cannot be used as a measure for cycle resolving. We therefore introduce new quantitative measures based on SCCs.

When examining the SCCs of the LOD-a-lot knowledge graph regarding popular relations, we observe two facts: 1) cycles of size two are very common across the graphs. When not nested into other cycles, they are SCCs with two nodes (SCCs of size two), which is the most common type of SCC. This suggests the ambiguity in definition and semantics of the relation; 2) there often exist a very big SCC that covers a majority of nodes involved in the SCCs. This is very different from synthetic models typically used in the evaluation of MWFAS algorithms. The following are measures on how much the SCCs are due to size-two cycles, and other complex nested cycles.

**Alpha measure.** Let $\alpha$ be the number of edges in cycles of size two divided by the number of all edges in its SCCs $\alpha = f_\alpha(G) = |E_{C2}|/|E^{SCC}|$. By definition, $f_\alpha(G) = f_\alpha(G^{SCC})$. This gives the fraction of edges that can be determined locally if given additional information (e.g. the reliability on each edge).

**Beta measure.** Remove all the cycles of size two from $G$ and obtain $G' = G \backslash E_{C2}$. The corresponding SCCs of $G'$ form a graph $G'^{SCC}$. Let $\beta$ be the number of edges in $G'^{SCC}$ divided by the number of all edges in $G^{SCC}$: $\beta = |E'^{SCC}|/|E^{SCC}|$. Similarly, we have $f_\beta(G) = f_\beta(G^{SCC})$. This measures the proportion of edges to make decisions on if all edges in size-two cycles are not involved in any SCC. In other words, it gives a measure of the fraction of edges in more complex nested cases.

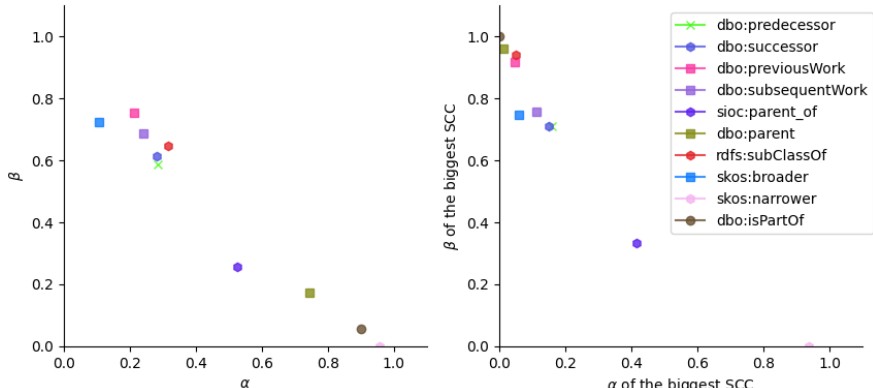

Fig. 3: The Alpha-Beta measures of representative relations

For the graph $G$ in Figure 2, $\alpha = 0.5$ and $\beta = 0.25$. As for its biggest SCC, $\alpha = 0.4$ and $\beta = 0.3$. Figure 3 reports on the $\alpha$ and $\beta$ values for the 10 selected relations in Table 1. The figure on the left illustrates the alpha-beta measure. In general, the greater $\alpha$ is, the more size-two cycles there are. The smaller $\beta$ is, the more likely it is to resolve the cycles by simply making decisions on the edges of cycles of size two (e.g. skos:narrower has $\beta = 0$). On the other hand, skos:broader and dbo:previousWork are examples with more complex cycles nesting into each other. An observation is that the tangent of a line crossing the origin and each point can indicate the hardness. This inspires the following definition.

**Gamma measure** : For an SCC $G$, the minimum fraction of decisions to be made to make $G$ cycle-free can be captured by $\alpha + \beta$. Note that an SCC $G$ gets harder when its $\beta$ is greater, or $\alpha$ is smaller. This can be captured by $\beta/\alpha$ or $\beta - \alpha$. To avoid cases where $\alpha = 0$ and to make $\gamma$ a term between 0 and 1, we use the latter and define $\gamma = f_\gamma(G) = (\alpha + \beta)(1 - \alpha + \beta)/2$. This gives a measure of the hardness to make an SCC cycle-free.

**Delta measure** : Note that using the Gamma measure, we can then estimate the effort required to make each of a graph's SCCs cycle-free. For a graph $G$ in general, $\delta = f_\delta(G) = \sum_{s \in \kappa(G)} f_\gamma(s) * |E|$. It is a sum over the $\gamma$ value multiplied by the number of edges of each of its SCCs.

Table 1 presents key information of the graphs of the 10 selected relations, together with the values of our metrics. For some graphs, a big proportion of the edges in SCCs are solely from its biggest SCC. The $\delta$ entries indicate that big graphs are not necessarily harder to resolve which is not captured by graph-theoretical measures. For example, the graph of skos:narrower is very big but has only 48 triples involved in cycles, making its $\delta$ value very small. These new measures provide a quantitative evaluation on the hardness of cycle resolving, help to study the nature of its complexity, and serve as references for the design of algorithms, choice of parameters as well as the sampling of data.

Table 1: Popular transitive and pseudo-transitive relations and their measures

| Relation ($R$) | $G_R$ | | SCCs of $G_R$ | | | | $G_R^B$ (the biggest SCC) | | | |
|---|---|---|---|---|---|---|---|---|---|---|
| | $|E_R|$ | $|V_R|$ | $|E_R^{SCC}|$ | $|V_R^{SCC}|$ | $|\kappa(G_R)|$ | $\delta$ | $|E_R^B|$ | $|V_R^B|$ | $\gamma$ | $\delta$ |
| skos:broader | 11.8m | 5.7m | 356.9k | 82.0k | 6.7k | 238.4k | 277.0k | 43.7k | 0.6 | 188.1k |
| rdfs:subClassOf | 4.4m | 3.6m | 1.4k | 837 | 196 | 961.05 | 780 | 301 | 0.9 | 730.4 |
| dbo:isPartOf | 1.0m | 408.3k | 4.7k | 3.8k | 1.5k | 312.8 | 60 | 29 | 1.0 | 60.0 |
| skos:narrower | 817.1k | 737.3k | 48 | 24 | 7 | 0.9 | 16 | 5 | 0.0 | 0.4 |
| dbo:previousWork | 551.2k | 550.1k | 10.6k | 8.4k | 1.5k | 8.0k | 710 | 469 | 0.9 | 639.2 |
| dbo:subsequentWork | 511.0k | 527.5k | 15.7k | 11.9k | 1.8k | 10.8k | 2.2k | 1.5k | 0.7 | 1.6k |
| dbo:successor | 440.7k | 417.3k | 60.2k | 38.0k | 5.8k | 36.2k | 12.5k | 5.9k | 0.6 | 8.4k |
| dbo:predecessor | 358.2k | 348.1k | 40.0k | 25.8k | 4.2k | 22.9k | 4.8k | 2.4k | 0.6 | 3.2k |
| dbo:parent | 105.8k | 97.0k | 9.7k | 4.3k | 921 | 1.9k | 1.5k | 979 | 0.9 | 1.4k |
| sioc:parent_of | 101.2k | 46.6k | 6.3k | 2.1k | 334 | 1.7k | 4.3k | 1.1k | 0.3 | 1.5k |

## 5   Algorithms

### 5.1   Algorithms for Cycle Resolving

We aim to design an algorithm that deals with knowledge graphs of (pseudo-) transitive relations that (i) does not rely on a ranking of nodes; (ii) captures the transitivity of relations; (iii) is capable of handling the complex structure resulted from large amount of nested cycles (graphs with high $\gamma$ values); (iv) is as conservative as possible in removing edges when resolving the cycles; and (v) is extendable to capture other logical and graph properties. The following section presents our algorithm with an evaluation.

We present Algorithm 1 as a general purpose cycle-resolving method for refinement. The algorithm exploits off-the-shelve technology for SMT solvers (Satisfiability Modulo Theories) [2]. The algorithm does not deal with reflexive edges as they can be processed trivially and in linear time. There are three main steps in the algorithm. We first compute the SCCs of the input graph (line 3). Then, we perform partitioning over big SCCs to a given bound $b_1$ (line 4). This is due to the limit of SMT solvers' capability to handle large amount of clauses. Finally, we sample some cycles and repeatedly call an SMT solver to identify edges to be removed (line 5-17). In the following, we discuss the strategies adopted to deploy it at web scale.

**Strategies for Graph Partition.** The graph partition problem is well-studied in graph theory. The minimum k-cut problem requires finding a set of edges whose removal partitions the graph to at least $k$ connected components. There exist efficient algorithms and open-source implementations. However, our experiments show that breaking an SCC $s$ into $k$ partitions directly results in a significant amount of edges being removed, whereas our goal is to be as conservative as possible in our repair strategy. For reducing the amount of edges to be removed, our **Strategy P1** partitions the graph into two subgraphs and then computes the SCCs. This process is repeated until each of the resulting SCCs are within the size bound $b_1$. For weighted graphs (see the next section for how weights are computed), we can adopt a **Strategy P2** by first removing the edges with the lower weights in size-two cycles, and then use Strategy P1.

---

**Algorithm 1:** General-purpose algorithm for cycle resolving

---

**1** **Input:** a graph $G$ with no reflexive edges, its weight function $f_w$ (optional), a bound $b_1$ for the number of maximum nodes for each SCC, and a bound $b_2$ for the number of hard clauses.

  **Result:** a cycle-free graph $H$ and a set of edges removed $A$.

**2** Initiate $A$ as an empty set;

**3** Compute the set of SCCs as $S$;

**4** Follow a graph partitioning strategy and reduce the size of $S$ to bound $b_1$ as $S'$ with removed edges collected and added to $A$;

**5** **while** $S'$ *is not empty* **do**

**6**   Initiate $S''$ as an empty set;

**7**   **foreach** $s \in S'$ **do**

**8**     Follow a sampling strategy, and obtain cycles $C$ from $s$ with $|C| < b_2$;

**9**     Initiate an SMT solver $o$;

**10**    Introduce to $o$ a set $P$ of propositional variable $p_e$ for each edge $e$ of $s$;

**11**    Encode cycles in $C$ as hard constraints in $o$;

**12**    Add to $o$ a clause of each variable $p_e \in P$ as a soft constraint with weight $f_w(e)$ if $f_w$ is present, otherwise 1;

**13**    Run the solver $o$ for optimal solution and decode the output model $m$;

**14**    From the model $m$, collect the edges $E$ to remove and let $A := A \cup E$;

**15**    Obtain a graph $s'$ from $s$ with $E$ removed;

**16**    Compute the SCCs $N$ of $s'$ and update $S'' := S'' \cup N$;

**17**  $S' := S''$;

**18** Remove edges $A$ from $G$ and obtain $H$.

---

**Strategies for Cycle Sampling.** The bottleneck for the earlier work in [18] was the combinatorial explosion when exhaustively listing all cycles of a graph. We therefore focus on sampling an amount of cycles in each iteration that balances the tradeoff between representative capacity and redundancy. **Strategy S1** focuses on the edges: choose a random edge $(s, t)$ in an SCC, then compute the shortest path from $(t, s)$. This forms a cycle. In total we collect $b_2$ such cycles. As an alternative strategy, we can adopt the **Strategy S2** that selects two nodes randomly and computes the shortest path from one to the other, and back.

**Resolving Cycles with an SMT solver.** This section gives details of the interaction with the SMT solver (line 9 and 13 in the algorithm). The SMT solver is used for two purposes: to satisfy all hard constraints and to satisfy the maximal amount of soft constraints.[4] The use of an SMT solver makes it possible to easily extend the current algorithm to weighted cases. In each iteration, for every $s \in S'$, we introduce a propositional variable $p_e$ for each edge $e$. When there is a cycle $v_1, \ldots .v_k$, we add a hard clause $[\neg p_{(v_1,v_2)} \vee \ldots \vee \neg p_{(v_{k-1},v_k)} \vee \neg p_{(v_k,v_1)}]$ to the SMT solver (accumulated in conjunction). The clause is satisfied when at

---

[4] A sub-optimal result is returned when an SMT solver reaches timeout.

least one of the $p_{i,j}$ is assigned False in the returned model of the solver, which indicates the removal of the edge $(i, j)$. To keep the maximal amount of edges, we add a soft clause $[p_e]$ for each edge $e$. The SMT solver performs a constrained optimisation process within a bounded time. The result of this is a near-optimal solution with the least amount of propositional variables set to False. From the model, we can retrieve edges to be removed to resolve all the encoded cycles. We repeat this process until all the SCCs are resolved and return the DAG and the removed edges. This approach can be easily extended to weighted cases, with the weight for each soft clause as the reliability for each edge.

### 5.2  Weights

Due to the logical foundation of knowledge graphs, repetition of statements is ignored because of the idempotency of the conjuction operator: $(\phi \wedge \phi) \leftrightarrow \phi$. Nevertheless, we believe that there is an important signal to be gained: the occurrence of the same triple in multiple knowledge graphs is an informal signal that multiple information providers have expressed support for. Thus, the chance that a statement is erroneous decreases with the number of knowledge graphs including this statement. Our algorithm takes the two kinds of weights for soft constraints (Algorithm 1, line 12). **Counted Weights**: the simplest way to obtain the weight of a triple is to count the number of occurrences across the graphs. The LOD-a-lot file consists of 650k datasets (graphs), making it feasible to compute such weights for popular relations; **Inferred Weights**: inspired by the observation and analysis in Section 4.1, we take advantage of the logical redundancy between implied properties to compute weights. If a triple (A `rdfs:subClassOf` B) is present in the integrated graph, and there is also an equivalence relation (A `owl:equivalentClass` B) or (B `owl:equivalentClass` A), then we make its weight 2 (i.e. we give more credence to (A `rdfs:subClassOf` B), otherwise 1). For `skos:broader`, we can take advantage of its inverse relation `skos:narrower`. If together with the triple (A `skos:broader` B) the triple (B `skos:narrower` A) exists in the dataset, then we assign weight 2 to the triple (A `skos:broader` B), otherwise 1. While counted weights always exist, inferred weights are more restricted and less common and requires some manual examination. Still, we experiment different weighting scheme for the sake of comparison in evaluation.

## 6   Experiments and Evaluation

### 6.1  Implementation and Parameter Settings

We implemented our algorithm[5] in Python. We adopt the Python binding of METIS[6], a graph partitioning package based on the multilevel partitioning

---

[5] `https://github.com/shuaiwangvu/Refining-Transitive-Relations`
[6] `https://github.com/inducer/pymetis`

paradigm providing quick and high-quality partitioning. We use Z3[7] as SMT solver [2], and the `networkx` package[8] for the handling of graphs and SCCs.

Based on some trial-and-error experience, in the following experiments we set $b_1 = 15,000$ (i.e. maximum size of an SCC before requiring graph partitioning), and apply Strategy P1 with $k = 2$. To balance the trade-off between efficiency against accuracy, we obtain $b_2 = 3,000$ clauses at most and set the time limit for the SMT solver to 10 seconds for each SCC. All experiments were conducted on a 2.2 GHz Quad-Core i7 laptop with a 16GB memory running Mac OS. All reflexive edges were eliminated in preprocessing.

## 6.2   Gold Standard

For evaluating hypothesis H2, we annotated a number of statements from the two most frequent (pseudo-)transitive relations (`rdfs:subClassOf` and `skos:broader`, according to Table 1).  For each relation, we have two gold standards. In the first gold standard **G1**, we randomly pick 500 edges from $E_R^{SCC}$. The second gold standard separates SCCs of two nodes (**G2-a**, 200 edges) from the rest (**G2-b**, 500 edges). When sampling **G2-b**, we first assign a number on each SCC according to their $\delta$-value and then sample the amount of edges assigned to each SCC randomly, thus providing an evaluation set for edges in complex nested cases. There are 1,199 unique edges in the gold standard of `skos:broader` with a total of 632 (52%) annotated 'remain' in contrast to 401 (33%) 'remove'. This analysis suggests that its under-specified definition caused confusion and subsequently resulted in a complex faulty graph structure. The great proportion of unknown entries for `rdfs:subClassOf` is discussed in Section 7.2.  The annotation process was conducted using the platform ANNit[9]. These gold standard datasets are online[10] together with detailed criteria, analysis and limitations. Its consistency was validated manually and by a Python script.

## 6.3   Efficiency Evaluation

In this section, we compare our refinement algorithm against other MWFAS algorithms. Table 2 presents the results of the number of edges removed for ten subgraphs of LOD-a-lot, both overall and within the SCCs. The highlighted cells show that our approach removes fewer edges during refinement. The result supports our Hypothesis H1. Both approaches are fast: general-purpose MWFAS algorithms take 2-12 seconds except KS, which may take up to 1 minute. Our algorithm takes 8-115 seconds except for `skos:broader`, which can take up to 8 minutes. Details of benchmarks are included in the repository of gold standard. The results in Table 2 are the best records of three runs. Finally, the results are validated to be free from SCCs except singletons.

---

[7] https://github.com/Z3Prover/z3

[8] https://networkx.github.io

[9] https://github.com/shuaiwangvu/ANNit

[10] https://zenodo.org/record/4610000

Table 2: The number of removed edges (with best results highlighted)

| Method | | skos:broader | rdfs:subClassOf | dbo:isPartOf | skos:narrower | dbo:previousWork | dbo:subsequentWork | dbo:successor | dbo:predecessor | dbo:parent | sioc:parent_of |
|---|---|---|---|---|---|---|---|---|---|---|---|
| BS | Overall | 1.1m | 4.3m | 18.8k | 57.7k | 113.4k | 107.1k | 85.5k | 67.8k | 16.9k | 6,568 |
| | SCCs | 327.0k | 1,160 | 3,291 | 33 | 7,542 | 11.1k | 43.7k | 28.9k | 7,282 | 6,026 |
| GRD | Overall | 493.1k | 25.3k | 2,175 | 3,462 | 11.9k | 12.4k | 24.8k | 17.2k | 5,270 | 1,867 |
| | SCCs | 356.9k | 430 | 2,153 | **20** | 2,555 | 3,776 | 17,679 | 11.8k | 3,937 | 1,639 |
| KS | Overall | 5.9m | 219.2k | 459.3k | 405.9k | 267.6k | 253.5k | 218.1k | 176.4k | 52.4k | 46.9k |
| | SCCs | 177.1k | 716 | 2,331 | 21 | 5,381 | 7,917 | 29.9k | 19.9k | 4,860 | 2,593 |
| DFS | | 125.6k | 529 | 2,286 | 21 | 2,309 | 3,558 | 17.0k | 11.4k | 3,943 | 1,913 |
| P1S1-unweighted | | **114.8k** | **330** | **2,143** | 22 | 1,953 | **2,952** | **13.3k** | **9,053** | 3,988 | **1,278** |
| P1S2-unweighted | | 144.0k | 360 | 2,169 | 22 | 2,161 | 3,253 | 20.1k | 10.1k | 4,151 | 2,681 |

## 6.4  Accuracy Evaluation

As for Hypothesis H2, we evaluate our algorithm's unweighted version against the two weighted versions (counted and inferred weights), as well as the MW-FAS algorithms. Table 3 presents the precision ($p$) and recall ($r$) as well as the number of removed edges ($|A|$) for skos:broader and rdfs:subClassOf. Each entry represents the average of three runs. Taking weights into account (especially counted weights) has a positive impact on precision while maintaining a similar recall. Our approach achieves the best precision among all methods while removing the least amount of edges. As for rdfs:subClassOf, the impact on precision can be positive but unstable due to the limits to be discussed in the next section. Overall, this evaluation gives positive support to our Hypothesis H2 and further enhances our conclusion for Hypothesis H1.

Table 3: Number of removed edges $|A|$, precision $p$, and recall $r$ for refinement

| Method | | skos:broader | | | | | | | rdfs:subclass | | | | | | |
|---|---|---|---|---|---|---|---|---|---|---|---|---|---|---|---|
| | $|A|$ | G1 | | G2-a | | G2-b | | $|A|$ | G1 | | G2-a | | G2-b | |
| | | $p$ | $r$ | $p$ | $r$ | $p$ | $r$ | | $p$ | $r$ | $p$ | $r$ | $p$ | $r$ |
| BS | 1.1m | 0.32 | **0.85** | 0.68 | **0.72** | 0.31 | **0.91** | 4.3m | 0.40 | **0.74** | 0.40 | **0.67** | 0.54 | **0.79** |
| GRD | 493.1k | 0.42 | 0.22 | 0.71 | 0.50 | **0.40** | 0.26 | 25.3k | 0.42 | 0.40 | 0.35 | 0.45 | 0.57 | 0.21 |
| KS | 5.9m | 0.33 | 0.52 | 0.74 | 0.53 | 0.28 | 0.46 | 2.1m | 0.38 | 0.43 | 0.43 | 0.55 | 0.54 | 0.53 |
| DFS | 125.6k | 0.35 | 0.37 | 0.68 | 0.49 | 0.34 | 0.34 | 529 | 0.43 | 0.42 | **0.49** | 0.63 | 0.55 | 0.29 |
| P1S1-unweighted | 114.8k | 0.32 | 0.26 | 0.73 | 0.52 | 0.30 | 0.28 | **330** | 0.50 | 0.51 | 0.45 | 0.57 | 0.40 | 0.11 |
| P1S2-unweighted | 142.6k | 0.32 | 0.35 | 0.73 | 0.52 | 0.31 | 0.37 | 350 | 0.49 | 0.44 | 0.45 | 0.57 | 0.58 | 0.15 |
| P1S1-inferred | 115.0k | 0.31 | 0.25 | 0.73 | 0.52 | 0.30 | 0.28 | **330** | 0.50 | 0.51 | 0.45 | 0.57 | 0.40 | 0.11 |
| P1S2-inferred | 143.8k | 0.33 | 0.38 | 0.73 | 0.52 | 0.30 | 0.36 | 354 | 0.49 | 0.46 | 0.45 | 0.57 | 0.60 | 0.14 |
| P2S1-inferred | 114.8k | 0.31 | 0.25 | 0.73 | 0.50 | 0.30 | 0.29 | **330** | 0.50 | 0.51 | 0.45 | 0.57 | 0.40 | 0.11 |
| P2S2-inferred | 142.7k | 0.33 | 0.35 | 0.73 | 0.52 | 0.31 | 0.37 | 356 | 0.50 | 0.47 | 0.45 | 0.57 | 0.58 | 0.15 |
| P1S1-counted | 95.4k | 0.40 | 0.33 | **0.78** | 0.55 | 0.34 | 0.26 | 335 | **0.53** | 0.49 | 0.45 | 0.57 | 0.67 | 0.16 |
| P1S2-counted | 98.3k | 0.42 | 0.38 | **0.78** | 0.55 | 0.34 | 0.28 | 354 | 0.51 | 0.45 | 0.45 | 0.57 | **0.70** | 0.20 |
| P2S1-counted | **93.4k** | 0.43 | 0.32 | **0.78** | 0.55 | 0.34 | 0.26 | 335 | **0.53** | 0.49 | 0.45 | 0.57 | 0.67 | 0.16 |
| P2S2-counted | 94.6k | **0.44** | 0.35 | **0.78** | 0.55 | 0.32 | 0.24 | 357 | 0.50 | 0.45 | 0.45 | 0.57 | 0.66 | 0.17 |

# 7   Discussion and Future Work

## 7.1   Summary

This paper presented a new algorithm for the refinement of transitive and pseudo-transitive relations in very large knowledge graphs. We employed an SMT solver in implementation and evaluated on 10 datasets and validated our Hypothesis H1. As a proof-of-concept, we extended our work to weighted knowledge graphs and evaluated on our gold standard. The results provided positive support for our Hypothesis H2 and we also showed that taking weights into account during refinement has a good potential.

## 7.2   Discussion

The graph of `rdfs:subClassOf` has 4.4 million triples, of which 1.4k are in SCCs. Only 17 triples have inferred weights greater than 1, while 292 triples have such counted weights. The `skos:broader` graph has 11.8 million triples, of which 265.9k are among SCCs. There are only 39 triples with inferred weights of 2 compared to 284.6k for counted cases. It is clear that far fewer triples are assigned inferred weights than counted weights, making it a less general weighting scheme. Table 3 shows that inferred weights have no significant impact on the results due to their small number. The following focuses on counted weights.

Figure 4 plots the frequency distribution of counted weights for both datasets. It shows a power law distribution for the weights of `skos:broader`, implying that some relation instances have been stated repeatedly across the web. This justifies the use of frequency of triples as a heuristic for reliability. In comparison, `rdfs:subClassOf` is less popular and its frequency distribution is less clear and thus less reliable for decision making.

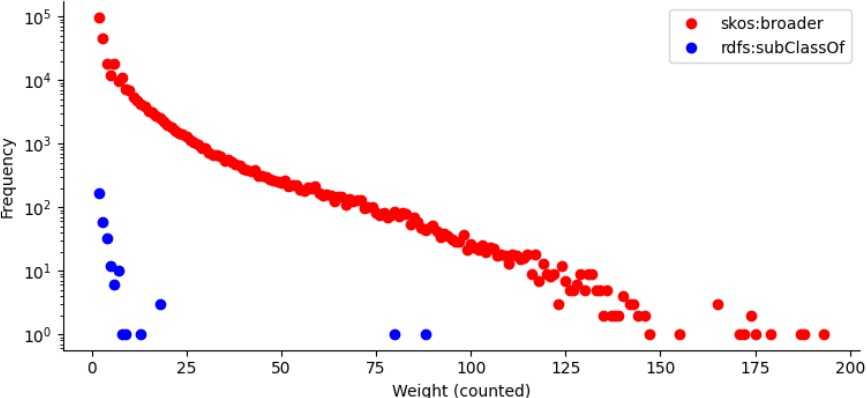

Fig. 4: The frequency distribution of counted weights in SCCs

Finally, the unstable result of `rdfs:subClassOf` is mostly due to the biggest SCC which has 780 edges, amounting to 52% of all the edges in SCCs. All these edges come from a single big faulty dataset, and are all annotated 'unknown'. This explains the big variance for precision and recall as in Table 3.

### 7.3   Limitations and Future Work

For Algorithm 1, at least one edge in each size-two cycle is removed. The algorithm can be further extended with some preprocessing where some size-two cycles are to be maintained by mapping both nodes to the same node in the corresponding homomorphic graphs.

Graph partition is an imprecise step in the algorithm. For example, among the 121.2k edges removed in the case of `skos:broader`, around 99.6k were identified during the graph partitioning step, amounting to 82.2%. Future work may optimise the parameters to balance the trade-off between accuracy and efficiency.

Figure 4 shows that the frequency distribution of `skos:broader` follows a power-law distribution. It can mislead the algorithm when an edge with a great weight is actually erroneous. Different weighting scheme can be explored such as that in [8]. Another possible way to improve the accuracy of weights and reduce the number of ties of weights in size-two cycles is by taking the reliability or centrality of sources into account as in [3] for example. General-purpose MWFAS algorithms can be adapted to their weighted cases for future evaluation.

Our recall is limited due to the small amount of edges removed. In Section 6.4, we restricted to these two relations due to their popularity and the great effort required for manual annotation. We plan to extend the gold standard to relations with different alpha-beta measures.

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
