# OpenReview forum: "Refining Transitive and pseudo-Transitive Relations at Web Scale"
_eswc-conferences.org/ESWC/2021/Conference/Research_Track — ESWC 2021 Research_

### Official Review · AnonReviewer4 · 2021-01-14
**Very generic approach but only scalable within bounds**

**Rating:** 1
**Confidence:** 2
**Impact:** 3
**Design And Technical Quality:** 3

**Review:**

This paper is about removing edges in a knowledge graph which would violate cyclic constraints
such as a cycle with skos:broder as the relation/edge label.
The authors present an approach based on a SMT(Satisfiability Modulo Theories) solver
which tries to remove the least number of edges to make the graph cycle free.

One of the positive points is that the whole algorithm is only graph based and can be applied to any relation.
Furthermore the approach is executed on real world data.
The main weak point is that the presented approach does not scale well, because as the authors wrote in section 5.1
`This is due to the limit of SMT solvers’ capability to handle large amount of clauses`.
Therefor two bounds b1 and b2 are artificially introduced to reduce the size of the SCC as well as the maximum number of nodes in a cycle.
The strategies P1 and P2 should enforce that the size of the SCC is smaller than b1.
P1 partitions the graphs as long as the constraints are met. But with this partitioning, edges are also removed.
And in this step, the number of removed edges are not minimized (at least from what I understand).
For the cycles sampling it is not clear why the shortet path and not just DFS is necessary to find a cycle.
Maybe the authors can write a bit more on this.

The relation work section is well written and covers many related approaches.
It is written that the DFS approach does not make any intelligent decision nor minimize the number of removed edges.
If the graph is weighted, the DFS approach may very well make an intelligent decision by removing the edge with the lowest weight in a found cycle.
This would be a stronger baseline for the weighted approach.
Furthermore there are approaches which are also graph based like [1]  which could be mentioned
and would further help e.g. the DFS to make better decision which edge to remove.

In section four, an analysis about the dataset is conducted.
It introduces measures (alpha, beta, gamma, delta) about how difficult a SCC is, when resolving cycles.
During reading I did not fully get why these measures needs to be presented (they do not help for the presented approach)
but I assume it is just some analyis about the data to see which cases are harder
(in case the authors need some more space, this section should be shortened).

The gold standard for the evaluation is made public and others can refer to it which is a positive point here.
What I miss in section 6.2 about the gold standard is the fact how it is generated
e.g. by how many people, what was the inter rater agreement, how was the task designed,
what help do the participants in the study get to analyze if an edge is correct or not, etc.
I also visted the link to ANNit but I did not find any further information.

The gold standard in zenodo has many unknowns (especially the ones for subclass; 425 unknown out of 500) as it can be seen from these tables:

rdfs:subclass:

remain | remove | unclear
--- | --- | ---
57 | 45 | 98
62 | 51 | 387
32 | 43 | 425

skos:broader:

remain | remove | unclear
--- | --- | ---
47  | 125 | 28
275 | 147 | 78
293 | 146 | 61

It would be good to know why there are so many unknowns and probably sample positives and negatives until for all test cases the sum of them are equal.
Then the tests will be comparable against each other.
And this might also be the reason why the results for rdfs:subclass are not as stable as for skos:broader.

In the parameter settings, the authors set the time limit for the SMT solver to 10 seconds for each SCC.
From what I understand, the SCCs are often small, but there are also a few huge ones.
Would it then not make more sense to calculate the timeout based on the number of nodes in the SCC?
And in more general: What happens if the algorithm runs in such a timeout (which happens mostly for the large SCCs)?
Is there a simple fall back or how are these cases handeled?

In the efficiency evaluation (section 6.3) the reader see some runtime measures
which shows that the skos:broader relation is more difficult.
What I don't see is the runtime of the DFS which should be much faster
(maybe put the runtimes in the table to allow easier comparison).

Some smaller points, typos etc:
- page 3 section 2.1: two times introduced (maybe rewrite it)
- page 3 section 2.1: and therefore cannot be applied on some of the cycles (maybe better: datasets) we present in this work
- page 5 figure 1: place the nodes of the graph variances at the same position as in the original graph to enhance readability
- page 13 table 2: dbo:predecessor -> approach P1S1-unweighted should be bold
- page 15 at the top: relation instances are 147:274 (for G1) -> the number should be switched (and better also include the unknowns here)
- the file names in the gold standard (zenodo) should be named equal to the GS names in the paper
- the github readme looks outdated because I can't find the file submassive-final.py (is it the main.py?)

[1] Breaking Cycles In Noisy Hierarchies. Jiankai Sun, Deepak Ajwani, Patrick K. Nicholson, Alessandra Sala, and Srinivasan Parthasarathy. 2017. In Proceedings of the 2017 ACM on Web Science Conference (WebSci '17). Association for Computing Machinery, New York, NY, USA, 151–160. DOI:https://doi.org/10.1145/3091478.3091495

I acknowledge I've read the rebuttals.

**Anonymity:**

Yes, I would like my review to remain anonymous.

**Reuse And Availability:**

4: High

**Strong Points:**

- released gold standard on zenodo
- released code on github
- approach is very generic and can be applied to any relation which is (pseudo) transitive
- new approach based on SMT solver to optimize the edge removal
- new measure for difficulty of cycle removal

**Subreviewer:**

I submitted this review.

**Weak Points:**

- the main approach is not really scaleable because the maximum number of nodes in a SCC is bound (by b1) as well as the maximum number of nodes in a cycle (bound by b2)
  - the authors provide some strategies for dealing with bound b1 (strategy P1 and P2) as well as cycle sampling S1 and S2, but this is just a workaround
- unclear how gold standard is created
- gold standard relatively small for subclass
- approach only a bit better than DFS (which could potentially be optimized)

---

> ### Author Rebuttal · Authors · 2021-01-30
>
> We thank the reviewer for taking the time to read our work and for the positive feedback and suggestions for improvement. Regarding the few aspects the reviewer addressed:
>
> Re scalability:
> Our approach takes a divide-and-conquer approach. The dividing part employs graph partitioning algorithms so the graph can be cut into two smaller sections by removing some relations, and then compute the SCCs of each. When reaching the bound b1, we are then able to perform more intellectual decisions using SMT. This two-step approach has been proved to scale for large graphs as we have shown. We tackled this scalability problem by repeatedly calling SMT over smaller graphs. But we agree that some more scalable algorithms could be developed to better deal with this issue.
>
> Re algorithm:
>
> >“Would it then not make more sense to calculate the timeout based on the number of nodes in the SCC?”
>
> Indeed it could be a more accurate way to fit the situation and that is a good way to use gamma. We will add the idea of dynamically setting the number of clauses using gamma in the future work.
>
> >“What happens if the algorithm runs in such a timeout (which happens mostly for the large SCCs)? Is there a simple fall back or how are these cases handled?”
>
> The algorithm of SMT returns a sub-optimal solution up to the timeout. We will add this clarification to the final version.
>
> Re measures:
>
> Indeed, as you write the purpose of the measure is “an analysis about the data to see which cases are harder”. The current version of the paper already explains that alpha “ gives the fraction of edges that can be determined locally if given additional information (e.g. the weight on each edge)” and that beta “gives a measure of the fraction of edges in more complex nested cases.” For gamma and delta we will add the following explanation:
>
> Gamma consists of two parts:
>
> Alpha +Beta: this shows the minimum portion of edges decisions have to be made
>
> (1 - Alpha + Beta)/2: this was used instead of (beta/alpha) to avoid the case where alpha is zero. It captures the difference of beta and alpha while maintaining the gamma value between 0 and 1.
>
> Delta is the total effort required to make the graph cycle free by summing all Gamma values multiplied by the number of edges.
>
>
> Re gold standard:
>
> Regarding the very many “unknown” entries for subclass: the observation is correct. As for subclass, there are two big SCCs (creationwiki and carleton) where the relations really do not make sense. But due to lack of further information, and the author of the carleton dataset passed away, we decided to let it remain undecided so it does not entirely shift the precision and recall values. This information will be added as a  note to the gold standard.
>
> We agree that the criteria of the gold standard could be presented more detailedly on github and we will do so soon.
>
> We thank the reviewer for the advice to adapt existing web-scale algorithms to weighted versions and compare against our method. This will be included as future work.
>
> Finally, we appreciate the pointers to references and the correction of typos.

---

### Official Review · AnonReviewer1 · 2021-01-14
**Interesting paper with some weaknesses**

**Confidence:** 4
**Impact:** 3
**Design And Technical Quality:** 3

**Review:**

## Quality

- the experiments are clearly described and the number of compared other systems provides a good basis for evaluation

- all artefacts are provided including code and data

- it is sometimes unclear what the criteria are for what the assumption defining "good" cycle elimination are.

  - in Section 1, as discussed below

  - in Section 2 for existing approaches, where it is often implicit

  - Section 2.1 gives measures for "hardness" of elimination cycles, but it is not clear what the measure for "good" is

  - on assumes that some minimality is desired, as cycle elimination in principle is always "simple" by deleting edges until no cycles remain (there is some evidence for that in Section 6.4 "Our approach achieves the best precision among all methods while removing the least amount of edges.")


## Clarity

- the paper is good to read throughout

- the introduction - while generally good - makes it not at all times clear what the exact problem, assumptions and quality metrics for the expected result are, making the reader guess more than necessary at this part

  - asymmetric and irreflexive relations are discussed, but it is not completely clear whether this is an example, an assumption or assertions (this is somewhat clarified in Section 4.1 later)

  - reliability is introduced as "we measure reliability by counting the number of occurrences across datasets" which is not completely clear

  - which "graph structural properties" does H1 ("By taking graph structural properties into account") refer to?


## Originality

- it is not completely clear whether related work is referenced ("In this work, we extend our analysis") - my assumption is that it is original


## Significance

- the paper clearly addresses an important topic

- not much argument is made of why the designed algorithm is particularly original or significant

- since it is not completely clear what the goal of "good" cycle elimination is, it is hard to say how significant the experimental results are


**Anonymity:**

Yes, I would like my review to remain anonymous.

**Rating:**

-1: Weak Reject

**Reuse And Availability:**

5: Very High

**Strong Points:**


- for significance, the paper clearly addresses an important topic

- for clarity, the paper is good to read throughout

- for quality, the experiments are clearly described and the number of compared other systems provides a good basis for evaluation

- for quality and clarity and significance, all artefacts are provided including code and data


**Subreviewer:**

I submitted this review.

**Weak Points:**


- for clarity, the introduction - while generally good - makes it not at all times clear what the exact problem, assumptions and quality metrics for the expected result are, making the reader guess more than necessary at this part

  - asymmetric and irreflexive relations are discussed, but it is not completely clear whether this is an example, an assumption or assertions (this is somewhat clarified in Section 4.1 later)

  - reliability is introduced as "we measure reliability by counting the number of occurrences across datasets" which is not completely clear

  - which "graph structural properties" does H1 ("By taking graph structural properties into account") refer to?

- for originality, it is not completely clear whether related work is referenced ("In this work, we extend our analysis") - my assumption is that it is original

- for quality and clarity, it is sometimes unclear what the criteria are for what the assumption defining "good" cycle elimination are.

  - in Section 1, as discussed above

  - in Section 2 for existing approaches, where it is often implicit

  - Section 2.1 gives measures for "hardness" of elimination cycles, but it is not clear what the measure for "good" is

  - one assumes that some minimality is desired, as cycle elimination in principle is always "simple" by deleting edges until no cycles remain (there is some evidence for that in Section 6.4 "Our approach achieves the best precision among all methods while removing the least amount of edges.")

- for originality and significance, not much argument is made of why the designed algorithm is particularly original or significant

- for clarity and significance, since it is not completely clear what the goal of "good" cycle elimination is, it is hard to say how significant the experimental results are

---

> ### Author Rebuttal · Authors · 2021-01-30
>
> We thank the reviewer for taking the time to read our work and for the positive feedback and suggestions for improvement. Regarding the few aspects the reviewer addressed:
>
> Regarding the quality of introduction and “good” cycle elimination.
>
> We agree that including an example in the introduction will help understanding the motivation of this paper and what “good” cycle elimination is. The following example showing 4 cycles will be included in the final version of the paper. It is a subgraph of the largest SCC of sko:sbroader with counted weights:
>
> <Numeral_system,  skos:broader, Integer>, with weight 1.
>
> <Numeral_system,  skos:broader, Numbers>, with weight 27.
>
> <Numbers,  skos:broader, Numeral_systems>, with weight 20.
>
> <Integers,  skos:broader, Numbers>, with weight 72.
>
> <Integers, skos:broader, Algebraic_numbers>, with weight 1.
>
> <Algebraic_numbers, skos:broader, Integers>, with weight 2.
>
> <Algebraic_numbers, skos:broader, Numbers>, with weight 41.
>
>
> From the weights we can tell that it is more certain that “Numbers” is a broader concept than “Integers”. However, it is very unlikely that “Numeral_systems” has a broader concept “Integers”.
>
> The example also indicates that it is impossible to construct a perfect gold standard. Take the edges between “Numeral_systems” and “Numbers” for example. When forming a link between pages, it is likely that a Wikipedia author would want to say that the study of Numbers includes its corresponding study of numerical systems, properties and many other aspects. While from a mathematical point of view, numbers are part of a specific numerical system. Thus leading to the confusion.
>
> We use two metrics to determine the value of a given solution. The first is "information loss", where we simply take the solution which removes the fewest cycles to be the best. This measure has the advantage that no annotation is required, but it is blind to whether the algorithm removes correct or incorrect edges. Our second measure is based on manually annotated data. Here we count a solution as good if many of the edges marked to be removed are indeed incorrect with respect to the manual annotations (high precision). In contrast, a bad elimination can remove many edges that are correct or follow from the semantics. This will be further clarified in the final version of the paper.
>
>
> Regarding clarity
>
> > asymmetric and irreflexive relations are discussed, but it is not completely clear whether this is an example, an assumption or assertions (this is somewhat clarified in Section 4.1 later)
>
> We consider it an assumption within the scope of this paper. We will make it clear in the final version.
>
> > reliability is introduced as "we measure reliability by counting the number of occurrences across datasets" which is not completely clear
>
> We have evaluated our approach on the LOD-a-lot, which is a crawl of the LOD cloud and consists of 650K datasets. A triple can appear in one or many datasets. We count the number of datasets it appears in and use that number as the weight of the edge. More means of defining weights can be explored as we suggested in the future work section. We will clarify this point.
>
> > which "graph structural properties" does H1 ("By taking graph structural properties into account") refer to?
>
> We are referring to the fact that cycles are nested into each other forming big and complex SCCs. An edge that participates in multiple cycles should be more likely to be removed. For the sake of efficiency, graph-theoretical methods often make decisions without considering the local neighbourhood. We will clarify this in the final version.
>
> > it is not completely clear whether related work is referenced ("In this work, we extend our analysis") - my assumption is that it is original
>
> The current paper does indeed go beyond our earlier work by being the first to work at web scale, by taking weights into account as a heuristic, showing how this can improve accuracy, and by measuring against a larger gold standard.
>
> > for originality and significance, not much argument is made of why the designed algorithm is particularly original or significant
>
> To the best of our knowledge this is the first work that uses SMT solvers for removing undesired cycles at web scale. The methods are proposed in a general and scalable manner, and can be applied for any arbitrary relation where acyclic graphs are desired.
>
>
> Finally, thank you for the correction of our typos, for listing the references and all the pointers. We will read the suggested papers in detail and finalise the reference list.

---

### Official Review · AnonReviewer2 · 2021-01-15
**Presents an interesting topic and breaks new ground, but some technical and experimental details need clarification**

**Rating:** 2
**Confidence:** 4
**Impact:** 4
**Design And Technical Quality:** 3

**Review:**

# Overview

The paper presents graph algorithms for refining (correcting) antisymmetric transitive and pseudo-transitive relations, i.e., properties whose graphs should not contain cycles. Such relations are identified from the LOD Laundromat collection, where the top 10 in terms of number of triples are extracted. Four measures are presented to quantify the level of difficulty in resolving the cycles of the graphs induced by such relations; primarily these measures are based on analyses of cycles of length 2. The measures for the graphs of the top 10 properties are then presented. Next the paper introduces a variety of algorithms for resolving cycles. The generic procedure relies on a Satisfiability Modulo Theories (SMT) solver, where the edge of each cycle is presented as an atomic literal, and the disjunction of their negation is passed as the constraint. The SMT solver should then find an approximately optimal solution (in terms of which edges to assign false/remove), also potentially taking into account weights on the edges represented as soft constraints. In order to reduce the computational load, strategies for partitioning the graph and sampling cycles are presented. Weights are presented based on counting in how many datasets it is found, and how through many "proofs" it can be derived. For evaluation purposes, the authors prepare a partial gold standard for two relations, labelling samples of triples as correct/incorrect/unknown. Results for the resolving the cycles of the dataset are then presented, taking into consideration the number of edges removed, along with the precision and recall with respect to the gold standard. Methods involving weights that count how many times an individual triple appears in a dataset seem to improve performance for one relation (skos:broader) but the results are less clear for the second relation (rdfs:subClassOf).

# Strengths (summary)

1. The proposed methods and line of research may ultimately lead to be a useful tool for improving the quality of knowledge graphs.

2. Overall I think the work is breaking new ground, and could encourage follow-up works.

3. The methods are proposed in a general manner and can be applied for arbitrary transitive/psuedo-transitive (acyclic) relations.

4. A gold standard is presented, which may also be reused in future research.

# Weaknesses (summary)

1. There are some technical aspects of the work that are left unclear.

2. The four measures appear a bit "ad hoc"; their value is unclear.

3. The results presented in the evaluation section are difficult to follow.

4. Details relating to the creation of the gold standard are vague.

5. Some related references are missing.

# Verdict

All in all, I think this is an interesting paper that deserves to be accepted. I hope/believe that the weaknesses mentioned above could be reasonably addressed as part of a camera ready version and should be weighed against the relative novelty of the paper and its contributions. (I provide more details on these weaknesses below.)

**Anonymity:**

No, I would like my review to be deanonymized.

**Reuse And Availability:**

5: Very High

**Strong Points:**

1. The paper presents methods for resolving cycles in what should be acyclic relations. Though perhaps a fraction of relations overall, this is an interesting problem, whose solutions (as proposed in this paper) could become an important part of a "toolkit" for refining knowledge graphs.

2. The work appears to me to be quite novel and introduces a number of new ideas. Overall I have the impression that this paper breaks new ground in terms of formalising an interesting problem, proposing initial techniques, formulating a gold standard, and establishing initial results.

3. While one or two papers have looked at similar problems, they have looked at such problems in the context of specific relations and datasets. The authors here present general-purpose methods that do not rely on relation- or domain-specific heuristics.

4. As part of "breaking new ground" the authors provide datasets, including a manually annotated gold standard. This is important for future research (and reproducibility).

**Subreviewer:**

I submitted this review.

**Weak Points:**

1. The paper covers quite a lot of ground, and perhaps as a result, some technical details are left vague. Also there are minor technical inaccuracies that should be resolved.

    - "$l_E : E \rightarrow \Sigma_E$" This definition maps each edge to a single label. For "a function that assigns to each edge $E$ a set of labels belonging to $\Sigma_E$" the definition should be $l_E : E \rightarrow 2^{\Sigma_E}$ where $2^S$ is used to denote the powerset of $S$.

    - "A walk in a directed graph G is a sequence of vertices and edges" The definition suggests it is only a sequence of vertices.

    - "We denote the set of such relations by $E_{C^2}$" Not clear what "such relations" means. Also a "relation" earlier was used to refer to an edge label/property rather than an edge/triple. Initially I had guessed that $E_{C^2}$ denotes self-loops that were previously mentioned, but I understand later that it refers to edges involved in cycles of length 2.

    - "$E_{C2}/|E|$" should presumably be $|E_{C2}|/|E|$

    - "By definition, $f_\alpha(G) = f_\alpha(G^{SCC})$." This does not seem to be the case unless I misunderstood: $E_{C2} = E_{C2}^{SCC}$, but $E \neq E^{SCC}$. In Figure 1, $|E_{C2}| = |E_{C2}^{SCC}| = 6$ and $|E| = 17$, $|E^{SCC}| = 12$. Thus $f_\alpha(G) = 6/17$ while $f_\alpha(G^{SCC}) = 6/12$.

    - In the definition of the gamma measure $\gamma(s)$, the variable $s$ is passed but never used. Likewise, in the delta measure, $\delta = \sum_{s \in \kappa(G)} \gamma * |E|$, the variable $s$ is quantified but not used on the right.

    - In Algorithm 1, it is unclear whether or not each SMT formula encodes one cycle or several cycles. I presume that it encodes several cycles as a trivial algorithm to resolve one cycle would be to just remove the lowest weighted edge. However, while the encoding for one cycle is presented, the encoding of multiple cycles is not (I presume it becomes a conjunction of the formulae for the individual cycles)?

    - It is not clear how the counts for “Inferred Weights” is defined formally (e.g., what entailment is considered, what) or how these counts are computed (e.g., is a reasoner used?). It seems though that ad hoc rules are used specifically for rdfs:subClassOf and skos:broader, which perhaps breaks the generality of the techniques used. (Perhaps this part could be removed and discussed more briefly, per the remarks in limitations.)

    - I found the description of the partitioning strategies to be a bit vague. I did not really understand what techniques are used. For example: "Strategy P1 repeatedly partitions the graph into a number of $k$ subgraphs". How is this partitioning conducted?

2. I don't really understand the purpose of the four metrics presented in Section 4.2. I vaguely understand that the idea is to characterise the difficulty of resolving cycles, but the measures themselves seem a bit "ad hoc" and their design principles are not made clear. In particular while the $\alpha$ and $\beta$ measures are quite intuitive, the $\gamma$ and $\delta$ measures are a bit obscure and not really motivated, just presented. It is not clear why $\delta$, for example, sums the $\gamma$ measures multiplied by the number of edges. More generally, it is not clear why so much emphasis is put on cycles of length 2 as a special case (when it would seem that cycles of length 3 would be easier to solve than cycles of length 30, for example). A more "direct" measure might be welcome, such as measures over the distribution of the number of (simple) cycles of length $n$ (though perhaps this would be computationally more intensive). Perhaps more generally, I don't see what would be lost by simply removing this section (and related discussion) from the paper.

3. I found the results of the experimental section difficult to follow. First, the section "Efficiency Evaluation" actually seems to focus on the number of triples removed, where runtimes are discussed only in passing. It is also not clear what to read into the number of triples removed: Are all methods guaranteed to output a DAG? Also is it not the case that removing many edges with low weight might be preferable to removing one (or few) edges with high weight?

4. The creation of the gold standard is not described in detail: it remains unclear what was the process of annotation in terms of who judged the edges, what criteria did they use, how many times each label is assigned, etc. Clarifying the process of annotation is important as clearly it is a non-trivial task to label these cases (which the paper itself acknowledges later). Having a quick look at the gold standard online, I found many cases where I would disagree. Taking a couple of examples from skos_broader_Gweighted_500:

    - http://dbpedia.org/resource/Category:Volcanoes_of_Africa	http://dbpedia.org/resource/Category:Volcanism_of_Africa	neither	remove	TBA
    - http://dbpedia.org/resource/Category:Landforms_of_the_North_Sea	http://dbpedia.org/resource/Category:North_Sea	neither	remove	TBA

    The labels indicate that skos:broader does not hold in either case nor in either direction. I would have labelled both of these as left<right without much doubt as clearly (for me at least) the left categories are proper sub-topics of the right categories. Of note that the skos:broader edges (involving DBpedia) ultimately come from Wikipedia categories, which (though obviously flawed) should reflect a manually-crafted consensus among editors; a judge marking this consensus as incorrect is actually contradicting one or more human editors who are probably more "involved" in the respective topics than the judge. I understand that these gold standards are created at considerable manual cost, but these sorts of ambiguities raise doubts for me about the reliability of the gold standards. A way to resolve this doubt (though perhaps “costly” in terms of manual effort) would be to have multiple evaluators, taking only those cases on which there was consensus, and also presenting a rater agreement to assess the level of subjectivity. (On a side note, I could not open some of the .gold files that begin with an understore, which appear to be hidden binary Mac files; please review. Perhaps these files are not important, I am not sure.)

5. There are a number of papers I think could be discussed in the related works (though they do not impact the novelty of the current work, overall).
    Of particular importance:

    - Marco Fossati, Dimitris Kontokostas, Jens Lehmann: Unsupervised learning of an extensive and usable taxonomy for DBpedia. SEMANTICS 2015: 177-184

    Algorithm 2 of the paper describes how they removed cycles from the Wikipedia category graph. I think it should be discussed.

    The authors also state that "we measure reliability by counting the number of occurrences across datasets. Associating each triple with such a weight induces a weighted knowledge graph. This feature has typically not been investigated by the community because the logical foundations of knowledge graphs dictate that repeated statements in datasets are redundant." I think this is a bit misleading in that there have been a number of works that have looked at similar measures of the reliability and importance of elements of RDF graphs, which are "boosted" by appearing in different sources. As an example, the following paper describes measures that rank elements of RDF graphs based on the sum of the centrality of the sources in which they appear:

    - Andreas Harth, Sheila Kinsella, Stefan Decker: Using Naming Authority to Rank Data and Ontologies for Web Search. International Semantic Web Conference 2009: 277-292

    We used a similar measure in previous work, in a more logical context, to decide which triples to remove in order to resolve inconsistencies during reasoning over multiple sources of data, which is a conceptually very similar problem to that addressed here:

    - Piero A. Bonatti, Aidan Hogan, Axel Polleres, Luigi Sauro: Robust and scalable Linked Data reasoning incorporating provenance and trust annotations. J. Web Semant. 9(2): 165-201 (2011)

    A benefit of using the sum of the centrality of the sources, rather than a simple count, is that not all documents are considered equal (e.g., triples from the FOAF spec that is linked from many locations will be weighted more highly than triples from an individual FOAF profile), and one can reduce the number of ties that would otherwise occur with simple counts. Though Figure 3 addresses this issue, there is still a high potential for ties in such a power law. In any case, either the original claim (which I believe to be misleading) could be revised, or some works using similar measures could be referenced. Also looking at weights based on a combination of the centrality of sources might be an interesting direction for future work.

---

# Minor comments
- "variances" -> "variants"

- "such as the widely used foaf:knows relation." Would be interesting to know why this happens. (I am almost sure it's not due to FOAF itself.)

- " in a graph[]"

- "when making cycle free" -> "when resolving cycles" or "when making a graph cycle free"

- ")[)], then we make its weight 2" Unbalanced parentheses.

- A standard measure of the "cyclicity" of graphs is treewidth. Unfortunately its computation is intractable, but there are implementations that work for many practical graphs.

- I think that Springer style requires table captions to go above the table.

- "the trade-off [between] accuracy against efficiency"

---

> ### Author Rebuttal · Authors · 2021-01-30
>
> We thank the reviewer for taking the time to read our work and for the positive feedback, spotting the typos, and suggestions for improvement.
>
> > the first four points (about typos and definition):
>
> We will fix the inconsistency between the notations in the final version and make the definition clearer and more accurate.
>
> > "By definition, fα(G)=fα(G_SCC)." This does not seem to be the case unless I misunderstood…”
>
> Since the edges of cycles of size two must be in the SCCs of G, it is equivalent to limit the scope to only the edges in the SCCs of G.
>
> >  In the definition of the gamma measure...
>
> Here s is an SCC, not an additional parameter. We will make it consistent with the definition of Alpha and Beta in the final version.
>
> > In Algorithm 1, it is unclear whether or not each SMT formula encodes one cycle or several cycles.
>
> Each cycle is encoded as a clause. All the clauses are loaded together in their CNF format (conjunction of the clauses). We will further update the last paragraph of section 5.1 to make this process clearer.
>
> >It is not clear how the counts for “Inferred Weights” is defined formally...
>
>  Inferred weight is defined as follows:
>
> a) For a triple <A, subclass, B>, if we also have <A, equivalentClassOf, B> or <B, equivalentClassOf, A>, then we assign the (inferred) weight to be 2. Otherwise, the (inferred) weight remains 1. The name “inferred” comes from its logical property: if <A, equivalentClassOf, B> or <B, equivalentClassOf, A>, then <A, subclass, B>.
>
> b) Similarly, for a triple <A, broader, B>, if we have <B, narrower, A>, then we assign an (inferred) weight to 2, otherwise 1. There are many cases of inverses and duals in the LOD cloud.
>
> No reasoner was used. We will include this explanation in the paper.
>
> > It seems though that ad hoc rules are used specifically for rdfs:subClassOf and skos:broader, which perhaps breaks the generality of the techniques used.
>
> The inferred weights are indeed not as general as the counted weights. However, we have observed that inverse relations exist in many cases. We will make both points clear in the final version.
>
> >... How is this partitioning conducted?
>
> We employed an efficient partitioning algorithm that is based on the multilevel graph partitioning paradigm providing quick and high-quality partitioning. This point will be further clarified where it was mentioned in Section 6.1.
>
> > About the description of alpha, beta, gamma and delta :
>
> We agree that the motivation for gamma and delta are less clear than for alpha and beta. We will therefore add the following to the final version:
>
> Gamma consists of two parts:
>
> Alpha +Beta: this shows the minimum portion of edges decisions have to be made
>
> (1 - Alpha + Beta)/2: this was used instead of (beta/alpha) to avoid the case where alpha is zero. It captures the difference of beta and alpha while maintaining the gamma value between 0 and 1.
>
> Delta is the total effort required to make the graph cycle free by summing all gamma values multiplied by the number of edges.
>
> The definition is based on simple cycles (of length n) because the following reasons:
>
> simple cycles do not capture the nature of this problem and do not make a difference between size-two cycles and bigger cycles, which is captured by Alpha and Beta.
>
> Since the cycles are nested into each other, the number of simple cycles can be massive. Thus computationally more expensive as the reviewer mentioned.
>
> Simple cycles of length n make it more of a local information but from our observation, there can be longer chains in an SCC. When the length is limited to n, these cycles will be ignored.
>
> The emphasis on size 2 cycles is due to its frequency in the SCCs: if there are many size two cycles, it suggests the ambiguity in semantics/definition. It is not only the most common kind of SCC, but also a very important aspect for complex nested cycles. SCCs/cycles with three nodes are much less common.
>
> The available space prohibits us to include more details on running time, variants of parameter settings, and its correspondence with delta. We will publish a table with further experimental results in the Github repository.
>
> The outputs are guaranteed to be DAGs.
>
> It is clear that taking weights into consideration will make it less likely to remove edges with greater weights. In fact, we noticed some edges are associated with a very large weight. In such case, if erroneous, it would be very hard for the SMT solver to identify. We included our thoughts on this in future work.
>
> Thank you for the suggestion regarding performing a new crowdsourcing-style evaluation and forming a new gold standard. We will include this suggestion in our future work.
>
> The files that you couldn’t open in our repo are just hidden binary files for macOS, that will be removed.
>
> The review is very detailed and very helpful!

---

> > ### Comment · AnonReviewer2 · 2021-01-31
> > **Response the rebuttal**
> >
> > I appreciate the clarifications of the authors.
> >
> > > Since the edges of cycles of size two must be in the SCCs of G, it is equivalent to limit the scope to only the edges in the SCCs of G.
> >
> > $f_\alpha(\cdot)$ uses the number of edges in the denominator. The number of edges is different for $G$ and $G_{SCC}$, and so while the numerator for $f_\alpha(\cdot)$ remains the same, the denominator changes? If I am mistaken, I am not sure why, so I would suggest to the authors to have a closer look at this and clarify or fix as appropriate.
> >
> > All of the other points addressed seem reasonable to me, though I perhaps have lingering doubts about the quality of the evaluation dataset used and the labelling process, which was also raised by other reviewers and not responded to in detail here. It would be better to use multiple judges and present inter-rater agreement (as other reviewers have also pointed out). The paper must mention how many judges were used, and if it were only one, this should be highlighted as a limitation in the paper.
> >
> > Overall I tend to believe my original recommendation is fair. The work has some limitations, but potentially breaks new ground.

---

### Official Review · AnonReviewer3 · 2021-01-15
**An efficient refinement algorithm that removes the least amount of links correcting transitivity-based relations cycles**

**Rating:** 2
**Confidence:** 2
**Impact:** 3
**Design And Technical Quality:** 4

**Review:**

A very well-thought paper that presents an efficient Web-scale KG refinement algorithm that removes the least amount of links to correct cycles in the graph structure based on OWL2 transitive properties.  The algorithm presented seems an improvement of previous work from the authors: reference [15].

Specific comments about the paper are presented below:
```{```
* Minor corrections:
	- pag. 08: the third line ends with double dots (..)
	- pag. 08: correct the layout of Table 1; there should be a column separation between col2 and col3.  Align to the center the text header.
	- pag. 09: for clarity, in Algorithm 1 line 10 (or before), define what is "P".
	- pag. 12: [6.2] Shouldn't the section title be "Gold Standard Datasets"?
	- pag. 12: [6.2] "As for G2-a," should be "As for G2-b"
	- pag. 12: page foot 9, remove the dot.
```}```

```
* Main evaluation (ranging from 0-100):
	+ Quality: 90
	+ Originality: 80
	+ Significance: 80

* Other aspects (ranging from 0-100):
	+ Clarity, illustration, and readability: 80
	+ Usefulness: 80
	+ Impression score: 90
```

---
# Post-rebuttal

I acknowledge I've read the rebuttals in the reviews.  Looking forward to reading the paper with the amendments.  Good job.

**Anonymity:**

Yes, I would like my review to remain anonymous.

**Reuse And Availability:**

4: High

**Strong Points:**

+ The paper is well-thought: its flow is smooth and the ideas follow a logical sequence easy to follow.
+ The paper presents two hypotheses.  In the last two sections, the paper presents supporting evidence and analysis regarding the veracity of both.
+ The cycle resolution with an SMT solver is a clever approach.
+ The algorithm's implementation along with the datasets are publicly available.


**Subreviewer:**

I submitted this review.

**Weak Points:**

+ [Section 4.2] Why gamma and delta were defined in that way? This should be more clear.
+ [Section 6.1] Needs more justification and, also, a more dynamic approach can be implemented in order to calculate proper values for the parameters based on the number of edges in the SCCs: (a) why 15k nodes?; (b) why at most 3k clauses and set the time limit to 10 seconds? Additionally, how edge cases are handled?
+ It would be interesting to have similar "Gold Standard" datasets for the other relations studied on the paper.  Also, the Gold Standard datasets should have the measures introduced in 4.2
+ The topic of accuracy and efficiency is complex.  It would be a good idea to have two specialized algorithms (and/or settings+parameters): one that focuses more on accuracy and another that focuses more on efficiency.  Any thoughts?

---

> ### Author Rebuttal · Authors · 2021-01-30
>
> We thank the reviewer for taking the time to read our work and for the positive feedback, spotting the typos, and suggestions for improvement.
>
> > Why gamma and delta were defined in that way? This should be more clear.
>
> Gamma consists of two parts:
>
> Alpha +Beta: this shows the minimum portion of edge decisions has to be made
>
> (1 - Alpha + Beta)/2: this was used instead of (beta/alpha) to avoid the case where alpha is zero. It captures the difference of beta and alpha while maintaining the gamma value between 0 and 1.
> Delta is the total effort required to make the graph cycle free by summing all Gamma values multiplied by the number of edges.
>
> We will include these phrases, and improve our motivation for defining these new measures  in the final version of the paper.
>
> > [Section 6.1] Needs more justification and, also, a more dynamic approach can be implemented in order to calculate proper values for the parameters based on the number of edges in the SCCs: (a) why 15k nodes?; (b) why at most 3k clauses and set the time limit to 10 seconds? Additionally, how edge cases are handled?
>
> As a first attempt for using SMT solvers to break cycles, these hyperparameters were set in an ad-hoc way. The initial parameters we started were with reference to the gamma value. We then performed trial-and-error evaluation for a good balance between accuracy and efficiency. We used our previous experience with SMT solvers when it comes to hyperparameter tuning. This explanation will be further clarified in the final version.
>
> > It would be interesting to have similar "Gold Standard" datasets for the other relations studied on the paper. Also, the Gold Standard datasets should have the measures introduced in 4.2
>
> We agree that extending the gold standard by (1) manually evaluating SCCs with different alpha/beta measures, and (2) considering additional relations would generalize better the results of this work. This limitation will be added to the future work section.
>
> > The topic of accuracy and efficiency is complex. It would be a good idea to have two specialized algorithms (and/or settings+parameters): one that focuses more on accuracy and another that focuses more on efficiency. Any thoughts?
>
> We agree that developing two specialised approaches would be beneficial, as some Linked Data applications might be better suited with a more conservative approach that removes a larger amount of possibly erroneous links. In this work, we only focused on finding a trade-off between efficiency and accuracy, but this point will be mentioned in the discussion section.

---

### Official Review · AnonReviewer5 · 2021-01-15
**The proposed approach is interesting, original and seems effective, but the experiments were done using one dataset only**

**Rating:** 1
**Confidence:** 3
**Impact:** 3
**Design And Technical Quality:** 4

**Review:**

This paper introduces an algorithm for refining (pseudo-)transitive relations in RDF graphs. The algorithm exploits graph structural properties for removing cycles and the reliability of triples, measured in number of occurrences across datasets, for identifying erroneous edges. The algorithm was evaluated using the LOD-a-lot dataset and two built gold-standard datasets. The experimental results show that the proposed algorithm is able to make graphs acyclic by removing less edges than graph theoretical methods and that has a better precision.

The addressed problem is significant and the proposed approach is interesting, original and seems effective.

I have two objections, though.

The first one is that the paper is hard to read. More examples would be welcomed. Also, the addressed problem should be better motivated. I miss examples in the introduction of undesired cyclic graphs of transitive relations and erroneous triples, and how the algorithm would resolve them. That would also help understanding the formal parts of the paper.

The second objection is that only one dataset was used in the experiments. In order to give more evidence of the effectiveness of the algorithm and, specially, to validate the second hypothesis and the proposed heuristics, more datasets should be used. For the second hypothesis, the chosen dataset is very specific as it only uses the relations rdfs:subClassOf and skos:broader.

**Anonymity:**

Yes, I would like my review to remain anonymous.

**Reuse And Availability:**

4: High

**Strong Points:**

- The addressed problem is significant
- The approach is interesting and original

**Subreviewer:**

I submitted this review.

**Weak Points:**

- The paper is hard to read
- The experiments were done using one dataset only

---

> ### Author Rebuttal · Authors · 2021-01-30
>
> We thank the reviewer for taking the time to read our work and for the positive feedback and suggestions for improvement. Regarding the two aspects the reviewer addressed:
>
> > The paper is hard to read. More examples would be welcomed. Also, the addressed problem should be better motivated.
>
> We will improve the readability and the motivation of the paper, and add as a motivating example a cyclic subgraph of the largest SCC of skos:broader, incorrectly modelling the relations between numbers, integers and numeral systems. (See the details in our answer to reviewer 4)
>
> > The experiments were done using one dataset only...
>
> We have evaluated our approach on the LOD-a-lot dataset, which is the result of merging more than 650K datasets crawled from the LOD cloud.
> For evaluating the efficiency of our approach (H1), we have used 10 different subgraphs of LOD-a-lot, with each subgraph representing a different (pseudo-)transitive relation.
> For H2, we have used two subgraphs of LOD-a-lot consisting of the two most frequent (pseudo-)transitive relations in this dataset (according to Table 1). We restricted the H2 evaluation to these two relations only, as it requires the construction of a gold standard of manually annotated triples for evaluating the accuracy of the approach. In addition, H2 also requires the presence of the statement’s frequency to evaluate their impact on accuracy. Therefore we are restricted to the use of web-scale datasets such as LOD-a-lot, for counting the statements’ occurrence across several RDF datasets.
>
> We will clarify these points in the final version. Moreover, we definitely agree that extending the gold standard by considering additional relations will generalize better the results of this work. This limitation will be added to the future work section.

---

### Decision · Program_Chairs · 2021-02-23

**Decision:**

Accept with shepherding

**Comment:**

All the reviewers agree that the authors addressed a significant problem interestingly and originally. During the discussion, the track chairs took into deep consideration the fact that the paper has the potential to breaks new ground, however, the author need to address important issues since the paper is accepted with shepherding which are listed in the following:
- additional papers to be discussed in the related work section
- add a motivating example
- some more clarification need to be addressed for:
-- technical aspects of the work
-- measures appear a bit "ad hoc";
-- results presented in the evaluation section
-- the creation of the gold standard